# CAMEO: CORRESPONDENCE-ATTENTION ALIGNMENT FOR MULTI-VIEW DIFFUSION MODELS

## ABSTRACT

We propose a novel framework designed to improve both the training efficiency and generation quality of multi-view diffusion models. While these models have emerged as a powerful paradigm for novel view synthesis (NVS) using their generative priors, they inherently lack geometry awareness as they have no 3D inductive bias. Moreover, they are typically trained with only a 2D denoising objective, leaving the learning process of geometric correspondence implicit and inefficient. In this work, we are the first to reveal that the 3D attention maps of these models exhibit an emergent property of geometric correspondence, attending to corresponding regions not only across reference views but also across target views. Furthermore, we observe that the model's generation quality strongly correlates with the alignment between its attention maps and geometric correspondence. Motivated by these findings, we introduce **CAMEO**, a simple yet effective training technique that directly supervises attention maps using geometric correspondence signals. Notably, supervising a single attention layer is sufficient to guide the model toward learning accurate correspondences, resulting in accelerated convergence and improved novel view synthesis performance. Applied to the CAT3D framework, the popular multi-view diffusion architecture, **CAMEO** accelerates convergence by 2.0× and achieves improved novel view synthesis quality. Code and weights will be publicly released.

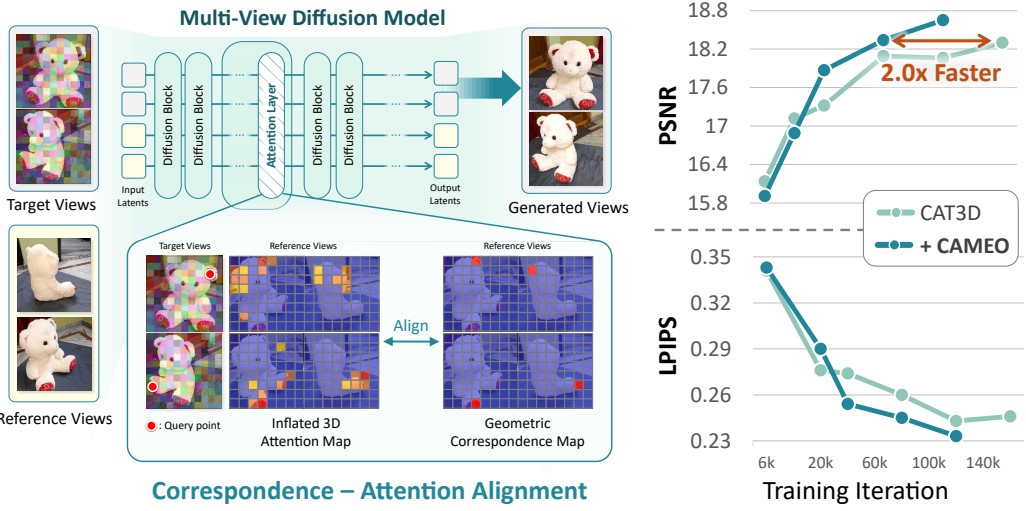

Figure 1: **Correspondence-attention alignment makes multi-view diffusion training efficient.** Our framework, **CAMEO**, explicitly aligns the multi-view diffusion models (Gao et al., 2024) with the geometric correspondence through a simple regularization. Model training becomes efficient and effective, achieves about 2.0× faster convergence than the vanilla model.

## 1 INTRODUCTION

Novel view synthesis (NVS) is the task of predicting images from unseen viewpoints given a set of reference views, while preserving geometric consistency and photorealistic appearance. Recent neural network-based NVS methods (Debevec et al., 1996; Szeliski & Golland, 1998; Mildenhall et al., 2021; Kerbl et al., 2023) rely on per-scene optimization, which requires dozens of input images, while generative NVS methods leverage the generation capabilities and 2D priors of generative models—particularly diffusion models (Ho et al., 2020; Rombach et al., 2022)—to synthesize novel views. These methods (Watson et al., 2022; Liu et al., 2023; Shi et al., 2023a; Gao et al., 2024; Cao et al., 2025; Szymanowicz et al., 2025; Zhou et al., 2025a) often extend the spatial attention mechanism of diffusion models into an inflated 3D spatial attention across all frames, enabling cross-view feature interaction for view-consistent image generation.

Despite their successes, training multi-view diffusion models remains challenging because they do not inherently possess 3D-aware inductive bias. Without explicit 3D representations (Mildenhall et al., 2021; Kerbl et al., 2023), the model must infer geometric correspondences across reference views to synthesize consistent novel views. However, training of these models relies solely on 2D diffusion loss (Ho et al., 2020), which solely aims for photometric similarity with GT image, which provides no direct guidance regarding multi-view geometric correspondences. As a result, learning novel view synthesis in multi-view diffusion models is a slow and inefficient process, with training effectiveness limited (Chan et al., 2023) as multi-view correspondence information has to be learned implicitly.

In this paper, we identify learning geometry as the central challenge in training multi-view diffusion models (Gao et al., 2024). We show that explicit supervision of correspondence information offers a simple and effective solution to this challenge. To this end, we propose a simple technique that directly supervises the multi-view diffusion models to infer geometric correspondences, leading to *faster training and enhanced generation quality*.

We investigate, for the first time, how geometric correspondences are represented within multi-view diffusion models. Inspired by recent works (Tang et al., 2023; Nam et al., 2025) showing that cross-attention layers in diffusion models capture correspondences, we analyze inflated 3D attention maps of multi-view diffusion models. Our findings reveal three key properties. (1) Despite being trained solely with a 2D denoising objective, multi-view diffusion models exhibit emergent geometric reasoning: their attention maps consistently capture geometrically corresponding regions across views, even when generating images from pure noise (Figure 2). (2) The alignment between attention maps and geometric correspondence maps—particularly in the final attention layer—shows a strong correlation with novel view synthesis quality (Figure 3). (3) This alignment becomes progressively sharper over the training process, indicating that geometric correspondence is acquired and refined by the model. These observations suggest that *geometric correspondence is inherently learned by multi-view diffusion loss, and it is explicitly represented within the attention layers of multi-view diffusion models.*

Building on these findings, we hypothesize that the training of multi-view diffusion models can be improved by providing explicit geometric supervision to their attention layers. To this end, we introduce **CAMEO** (Correspondence–Attention Alignment for Multi-view Diffusion), a simple yet effective technique that distills geometric correspondences into the model's attention layers. We construct a correspondence map as the supervision for geometry and align the model's attention map with it during training. Remarkably, aligning attention of a single layer in the final 3D attention block proves sufficient to improve learning. This explicit geometric supervision guides the model to learn more precise geometric correspondences, resulting in accelerated convergence and improved novel view synthesis quality. Note that our method is model-agnostic and can be integrated into any multi-view diffusion model architecture that employs an inflated 3D attention mechanism.

To evaluate the effectiveness of our method, we implement CAMEO on a representative multi-view diffusion model, CAT3D (Gao et al., 2024), We conduct comparisons on RealEstate10K (Zhou et al., 2018) dataset. On average, CAMEO reduces the training time by 2.0×, 80k steps to surpass a PSNR of 18.3, whereas vanilla model requires 160k steps.

The main contributions of this paper are as follows:

- We analyze how geometric correspondences emerge in the attention maps of multi-view diffusion models and demonstrate their strong correlation with NVS performance.

- We propose **CAMEO**, a simple attention-alignment method that explicitly supervises geometric correspondences in multi-view diffusion models.

- We validate CAMEO on CAT3D (Gao et al., 2024) baseline, showing that it accelerates training by $2.0\times$ and outperforms both the baseline and recent feature-alignment methods on the RealEstate10K benchmark.

## 2 RELATED WORK

**Diffusion models for novel view synthesis.** Diffusion models (Ho et al., 2020; Rombach et al., 2022) have emerged as powerful generative priors for novel view synthesis (NVS), moving beyond traditional geometry-based methods (Mildenhall et al., 2021; Kerbl et al., 2023). While early approaches (Watson et al., 2022; Liu et al., 2023; Shi et al., 2023a) formulated NVS as a conditional image-to-image translation task, recent multi-view diffusion models (Gao et al., 2024; Cao et al., 2025; Szymanowicz et al., 2025; Zhou et al., 2025a) jointly generate sets of geometrically consistent views. These methods extend 2D latent diffusion models by inflating the self-attention mechanism into inflated 3D attention, enabling information exchange across reference and target views.

Despite their impressive results, these models lack explicit 3D inductive bias and must learn scene geometry implicitly (Chan et al., 2023). Their training is guided solely by 2D supervision via the denoising diffusion loss, without any direct geometric signal. As a result, they require large-scale training data and considerable computational resources (Cao et al., 2025).

**Attention map alignment for knowledge distillation.** Attention alignment is widely used in knowledge distillation, particularly in Transformer-based models, to transfer structural knowledge from teacher to student by matching attention distributions (Jiao et al., 2019; Sun et al., 2020).In vision tasks, aligning attention maps has been shown to improve generalization in self-supervised transformers (Wang et al., 2022), enhance spatial consistency in dense prediction (Ji et al., 2021), and remain effective even without feature-level supervision (Li et al., 2024). Recent work extends attention distillation to generative settings, embedding supervision into the sampling process of diffusion models for style and semantics transfer (Zhou et al., 2025b). Our work builds on this foundation by aligning multi-view diffusion model's attention map with geometric correspondences to inject 3D supervision into the model.

**Representation alignment.** Aligning intermediate representations with pre-trained encoders has emerged as an effective strategy for improving diffusion model training. REPA (Yu et al., 2024) introduces feature-level supervision by distilling semantic features from DINOv2 (Oquab et al., 2023) into early layers of a Diffusion Transformer (Peebles & Xie, 2023), accelerating convergence and enhancing semantic structure. This idea is extended temporally by Video-REPA (Zhang et al., 2025), which enforces inter-frame feature consistency in video diffusion models, yielding improved temporal coherence. These methods highlight the benefit of structured feature alignment in both static and sequential generative tasks.

## 3 METHOD

### 3.1 BASELINE ARCHITECTURE

The goal of novel view synthesis (NVS) is to generate $M$ target images $\{\mathcal{I}_i^{\text{tgt}}\}_{i=1}^{M}$ for a target camera poses $\{\pi_i^{\text{tgt}}\}_{i=1}^{M}$, given a set of $N$ reference images $\{\mathcal{I}_i^{\text{ref}}\}_{i=1}^{N}$ along with their corresponding camera poses $\{\pi_i^{\text{ref}}\}_{i=1}^{N}$. This task requires reasoning about the underlying 3D scene structure to generate target images aligned with the target camera, ensuring consistency with reference images.

Existing generative NVS methods leverage multi-view diffusion models to learn joint distribution of novel views conditioned on one or more posed reference images (Gao et al., 2024; Shi et al., 2023b). They employ an inflated 3D self-attention mechanism to learn 3D-aware features across all views to generate geometrically consistent views. Specifically, it extends the standard 2D attention in image

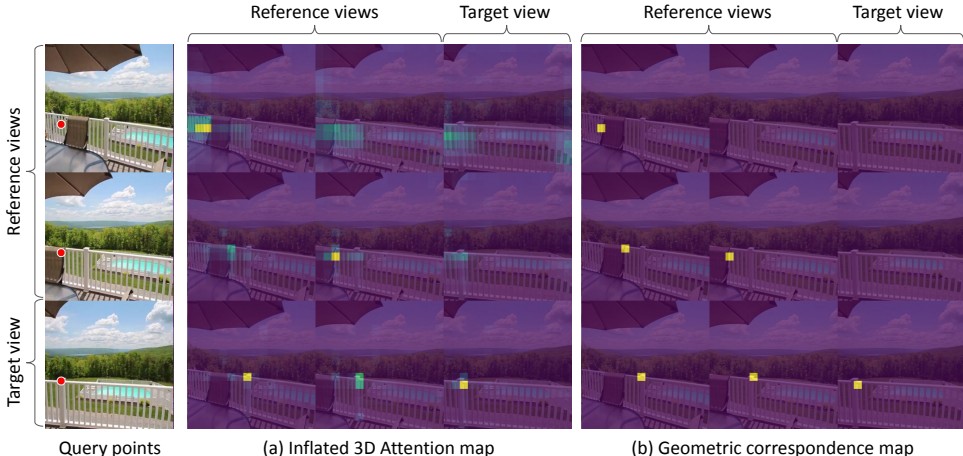

Figure 2: **Attention maps in multi-view diffusion models align with the correspondence map.** (a) Attention map from CAT3D Gao et al. (2024) at layer $\ell = 10$. (b) Geometric correspondence map. Even without explicit supervision, the model learns to attend to its geometric counterpart across views. Given the same query pixels, both (a) and (b) exhibit similar correspondence patterns.

diffusion models by concatenating the token sequences from each view, enabling interactions both within and across views.

At each attention layer of the model, features from each of the $F(= N + M)$ images are projected into query ($Q_i$) and key ($K_i$) matrices, each of size $\mathbb{R}^{hw \times d}$. These are then concatenated along the spatial axis using $\mathrm{Concat}(\cdot)$, which stacks tokens from all $F$ views into a single sequence, with $N$ reference views followed by $M$ target views. This produces the final matrices $Q$ and $K$:

$$Q = \mathrm{Concat}\left(Q_1, \ldots, Q_N, \ Q_{N+1}, \ldots, Q_F\right) \in \mathbb{R}^{Fhw \times d},$$
$$K = \mathrm{Concat}\left(K_1, \ldots, K_N, \ K_{N+1}, \ldots, K_F\right) \in \mathbb{R}^{Fhw \times d}. \tag{1}$$

### 3.2 ANALYSIS

We investigate how multi-view diffusion models encode 3D geometry solely through the implicit supervision of the diffusion loss. In particular, we analyze whether the inflated 3D self-attention mechanism learns to capture inter-view correspondences between target and reference images. To evaluate this, we obtain geometric correspondence map from point map and measure the similarity between the model's attention maps and the correspondence maps.

**Geometric correspondence.** Given a set of images $\{\mathcal{I}_i\}_{i=1}^F$ and their corresponding pixel-wise 3D point maps $\{\mathcal{X}_i\}_{i=1}^F$, where $\mathcal{I}_i \in \mathbb{R}^{H \times W \times 3}$ and $\mathcal{X}_i \in \mathbb{R}^{H \times W \times 3}$, we construct geometric correspondences between views by comparing the 3D location of pixels across images. Specifically, for a pixel $p = (x, y)$ in image $\mathcal{I}_i$, we aim to find its geometric correspondence $p^*$ in another image $\mathcal{I}_j$ ($j \neq i$). We first obtain the 3D coordinate $\mathcal{X}_i(p)$ from the point map of $\mathcal{I}_i$. Then, among all pixels $(u, v)$ in $\mathcal{I}_j$, we find the one whose 3D point in $\mathcal{X}_j$ is closest to $\mathcal{X}_i(p)$. Formally,

$$p^* = \arg\min_{(u,v)} \|\mathcal{X}_i(p) - \mathcal{X}_j(u, v)\|_2. \tag{2}$$

The correspondence map from pixel $p$ in $\mathcal{I}_i$ to image $\mathcal{I}_j$ is defined as a one-hot spatial map $\mathcal{P}_{i \to j}(p) = \mathbf{1}[(u, v) = p^*]$, where the location corresponding to the matched pixel $p^*$ is set to 1 and all other entries are set to 0. These maps serve as ground-truth correspondence for evaluating how well the attention maps of multi-view diffusion models capture geometric correspondences. An example of such a map is shown in Figure 2 (b). The pixel-wise 3D point maps used to derive these correspondences were obtained using an off-the-shelf geometry estimation model (Wang et al., 2025).

**Correspondence in multi-view diffusion models.**  For our analysis, we examine CAT3D (Gao et al., 2024) (see Appendix B for a detailed CAT3D architecture). To quantify the capability of the model's attention captures geometric correspondences, we compare the predicted attention maps with the correspondence maps. First, for a given query pixel location $p$ of image $\mathcal{I}_i$, we identify its corresponding token location $p_t$ in the latent space. We extract the query embedding $Q_i(p_t) \in \mathbb{R}^{1 \times d}$, and compute its similarity with key tokens $K_j \in \mathbb{R}^{hw \times d}$ from reference view $j$. This gives us a attention map between paired view $i$ and $j$ at layer $\ell$:

$$\mathcal{A}_{i \to j}^\ell(p_t) = \mathrm{Softmax}\left(\frac{Q_i(p_t)K_j^\top}{\sqrt{d}}\right) \in \mathbb{R}^{hw}, \tag{3}$$

where $i \to j$ indicates that the query is from view $i$ and the key is from view $j$, and $hw$ denotes the spatial resolution in the latent space. Next, we interpolate the correspondence map $\mathcal{P}_{i \to j}(p)$ to match its spatial dimension with $\mathcal{A}_{i \to j}^\ell$. The final correspondence map is defined as $\tilde{\mathcal{P}}_{i \to j}(p_t)$.

**Analysis metric.**  We define the alignment error at layer $\ell$ as the expected cross-entropy $\mathrm{CE}(\cdot)$ between the predicted attention map and the correspondence map over all valid view pairs $(i, j)$ and all query tokens $p_t$:

$$\mathcal{E}_{\mathrm{align}}^\ell = \mathbb{E}_{(i,j),\, p_t}\left[\mathrm{CE}\left(\mathcal{A}_{i \to j}^\ell(p_t),\, \tilde{\mathcal{P}}_{i \to j}(p_t)\right)\right], \tag{4}$$

This metric reflects how well the model's attention aligns with geometric correspondences. The final result is obtained by averaging $\mathcal{E}_{\mathrm{align}}^\ell$ across 200 validation samples from the RealEstate10K (Zhou et al., 2018) dataset. Further details on the analysis setup and supplementary results can be found in Appendix C.

**Attention maps in multi-view diffusion are already aligned with correspondence maps.**  As shown in Figure 2, for a selected query point, the attention map accurately attends to its geometric counterpart. The attention map is already aligned with the geometric correspondence map without any direct supervision. Notably, we observe that this property emerges in the deeper layers (see Figure 12), which are more closely tied to the final output. Layer $\ell = 10$ consistently produces aligned attention maps, while the earlier layers lack clear correspondence patterns. This suggests that learning accurate correspondences becomes critical at later stages, where they directly contribute to generating coherent novel views.

**Final layers strongly link attention alignment with generation quality.**  We perform a per-layer analysis to assess whether better attention alignment corresponds to improved generation quality. Specifically, we compute the cross-entropy alignment error $\mathcal{E}_{\mathrm{align}}^\ell$ at each layer $\ell$ and evaluate its correlation with image quality metrics—PSNR, SSIM, and LPIPS—using the Pearson correlation coefficient (PCC). PCC values range from $-1$ to $1$, where values closer to $1$ indicate that lower alignment error is strongly associated with higher quality. For consistency in the analysis, PSNR and SSIM—metrics where higher is better—had their signs inverted.

As shown in Figure 3 (a), layers $\ell = 10, 11, 12$ exhibit strong positive correlation with image quality metrics, while others show weak or no correlation. This indicates that in the final layers, the ability of attention maps to localize geometrically accurate correspondences is closely linked to generation quality. These findings offer insight into how internal attention behaviors contribute to the output quality in multi-view diffusion models.

**Final layers are more capable of learning geometric correspondences.**  In Figure 3(b), we show how the alignment between CAT3D's attention maps and the correspondence maps evolves during training. Notably, layer $\ell = 10$ exhibits the strongest alignment, starting with the lowest alignment error and improving steadily throughout training. In contrast, layers $\ell = 2, 4, 7$ remain relatively stable, showing little change in alignment. Layer $\ell = 6$ shows the weakest alignment. It begins with the highest error, improves slightly during training, but remains higher than all other layers. This analysis suggests that the final layers in U-Net are more capable of learning geometric correspondences.

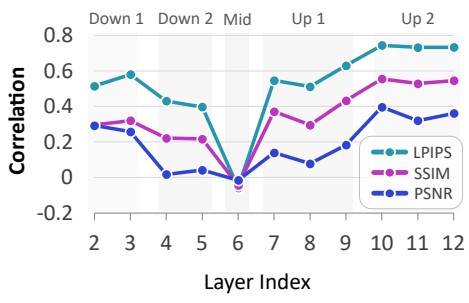
(a) Correlation between $\mathcal{E}_{\text{align}}^{\ell}$ and metrics

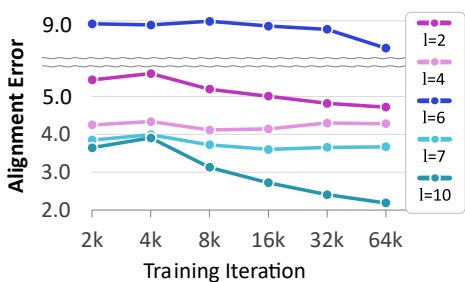
(b) $\mathcal{E}_{\text{align}}^{\ell}$ during training at different layers

Figure 3: **Alignment behavior for a multi-view diffusion model (Gao et al., 2024)**. We investigate the attention map alignment across layers in multi-view diffusion model (a) Final layers ($\ell = 10, 11, 12$) show a strong correlation between reduced alignment error and generation quality. (b) Final layers $\ell = 10$ reduce alignment error significantly during training, while others remain nearly constant. We show the first layers of each block ($\ell = 2, 4, 6, 7, 10$), as others show similar trends; full results are in Appendix C.2.

### 3.3 CORRESPONDENCE-ATTENTION ALIGNMENT

Motivated by these findings, we propose a correspondence-attention alignment, which explicitly supervises attention maps to match geometric correspondence maps, enabling the model to generate more coherent and geometrically accurate novel views.

Given $N$ reference images and $M$ target images, where $F = N + M$, each cross-attention layer $\ell$ of the diffusion model computes inflated 3D map $\mathcal{A}^{\ell} \in \mathbb{R}^{(Fhw) \times (Fhw)}$. It encodes geometry correspondence across all views, where $\mathcal{A}_{i \to j}^{\ell}(p_t) \in \mathbb{R}^{h \times w}$ is a probability map in the view $j$ for the query point of $p_t$ in the view $i$. To facilitate learning of implicit 3D geometry, we inject supervision by providing a geometric correspondence map during training. We first construct correspondence map $\tilde{\mathcal{P}} \in \mathbb{R}^{(Fhw) \times (Fhw)}$, where $\tilde{\mathcal{P}}_{i \to j}(p_t) \in \mathbb{R}^{h \times w}$ is one-hot map indicating corresponding point in 3D space of location $p_t$ in the view $i$. Then we regularize the diffusion model in the training to align $\mathcal{A}_{i \to j}^{\ell}(p_t)$ and $\tilde{\mathcal{P}}_{i \to j}(p_t)$ for all view pairs $i, j \in [1, F]$ and query points $p_t$, guiding the model to learn multi-view consistency.

Since some regions are not visible in any other view, the raw correspondence map $\tilde{\mathcal{P}}$ can contain noisy or ambiguous entries. To mitigate this, we introduce a binary mask $\mathcal{M}_{i \to j} = f_{\tau}(\tilde{\mathcal{P}}_{i \to j}, \tilde{\mathcal{P}}_{j \to i})$, where the function $f_{\tau}$ checks cycle consistency of correspondences. Specifically, for a query index $p_t$, we find its corresponding location $p_t^*$, and its reverse match $\hat{p}_t$. Then the mask $\mathcal{M}_{i \to j}(p_t) = 1$ only when the distance of $p_t$ and $\hat{p}_t$ in the feature space is lower than threshold $\tau$.

Given the subset of inflated 3D attention layers $\mathcal{S}$ in the diffusion model, we define the attention alignment loss as:

$$\mathcal{L}_{\text{CAMEO}} = \mathbb{E}_{\ell \in \mathcal{S}, \ (i,j), \ p_t} \left[ \mathcal{M}_{i \to j}(p_t) \odot \text{CE} \left( \mathcal{A}_{i \to j}^{\ell}(p_t), \ \tilde{\mathcal{P}}_{i \to j}(p_t) \right) \right], \tag{5}$$

where $\text{CE}(\cdot)$ is the cross-entropy loss and $\odot$ is element-wise multiplication.

Our final training objective combines the standard denoising score matching loss $\mathcal{L}_{\text{denoise}}$ used in diffusion models (Ho et al., 2020) with the proposed correspondence-attention alignment loss:

$$\mathcal{L}_{\text{total}} = \mathcal{L}_{\text{denoise}} + \lambda \cdot \mathcal{L}_{\text{CAMEO}}, \tag{6}$$

where $\lambda$ is a hyperparameter. For the effectiveness, we employ a projection head on the attention logits before softmax, using a simple multilayer perceptron (MLP).

## 4 EXPERIMENTS

We evaluate the effectiveness of **CAMEO** by addressing the following key questions:

- Can CAMEO accelerate the training of multi-view diffusion models?

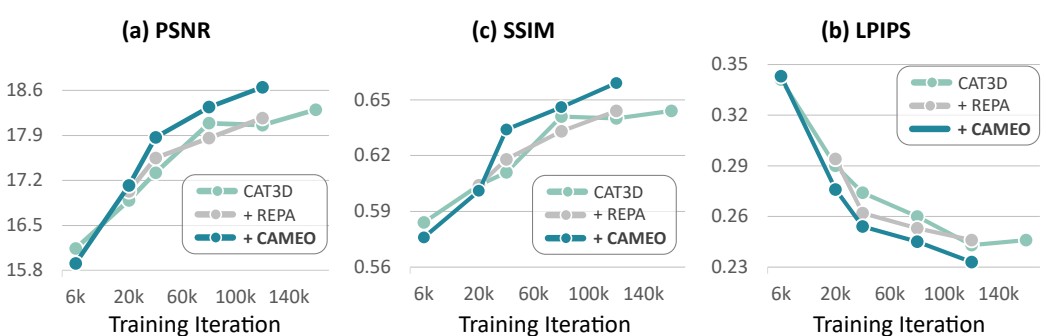

Figure 4: **The relative improvements of CAMEO over vanilla model.** We report $2.0\times$ times speed up in PSNR, LPIPS, and SSIM.

- Does CAMEO improve the quality of novel view synthesis?
- Does CAMEO remain effective under out-of-distribution (OOD) settings, demonstrating generalization beyond the training distribution?

## 4.1 IMPLEMENTATION DETAILS

**Model.** We adopt CAT3D as the baseline multi-view diffusion model for our experiments. Since the official implementation of CAT3D is not publicly available, we employ the re-implementation provided by MVGenMaster (Cao et al., 2025). Following prior works (Liu et al., 2023; Gao et al., 2024), we initialize all models from pretrained text-to-image diffusion parameters (Rombach et al., 2022). Additional details about the model are provided in the Appendix B.

**Dataset.** We use RealEstate10K dataset (scene-level) (Zhou et al., 2018) for training at a resolution of $512\times512$. Each training sample consists of 4 input views, where 1–3 views are randomly masked as target views, and the rest as references. For the main evaluation, we randomly sample 280 scenes from the RealEstate10K test set (Zhou et al., 2018) and evaluate performance under both 1-to-3 and 2-to-2 view settings, covering a diverse range of camera poses. For out-of-distribution (OOD) evaluation, we use the validation split of DTU validation set (object-centric) (Jensen et al., 2014), processed by (Chen et al., 2024), and conduct evaluation under 2-to-2 view setting.

**Diffusion Setup.** Following (Gao et al., 2024; Cao et al., 2025), we apply classifier-free guidance (CFG) (Dhariwal & Nichol, 2021) training by randomly dropping camera condition with probability of 0.1. At inference, we use DDIM sampler (Song et al., 2020) with 50 sampling steps and CFG with weight of 2.0.

**Training.** For fair comparison, we keep the batch size to 6 and train models with AdamW optimizer (Loshchilov & Hutter, 2019), adopting a fixed learning rate of 2.5e-5 and a weight decay of 0.01. All experiments are conducted on 2 NVIDIA A100 (40GB) GPUs.

## 4.2 EXPERIMENTAL RESULTS

We apply CAMEO to layer $\ell = 10$, which demonstrates a strong relationship with geometric correspondence (Figure 2), with the loss weight of $\lambda = 0.02$. We also include REPA (Yu et al., 2024), a recent feature alignment method that accelerates diffusion model training.

**Training Efficiency.** To investigate how CAMEO influences the training dynamics of multi-view diffusion models, we compare CAMEO with the baseline at intermediate training steps. As shown in Figure 4, our method achieves substantially faster convergence-performance than the baseline. Specifically, CAMEO reaches a PSNR above 18.3 at 80k iterations, whereas the baseline requires 160k iterations to achieve the same performance — corresponding to a $2\times$ acceleration. These results demonstrate that correspondence–attention alignment enables more efficient learning of geometric structure in multi-view diffusion models.

**Novel View Synthesis Quality.** The benefits of CAMEO extend beyond training efficiency to improvements in the final quality of novel view synthesis. As shown in Table 1, CAMEO surpasses both the baseline and REPA in overall performance. Furthermore, Figure 5 demonstrates

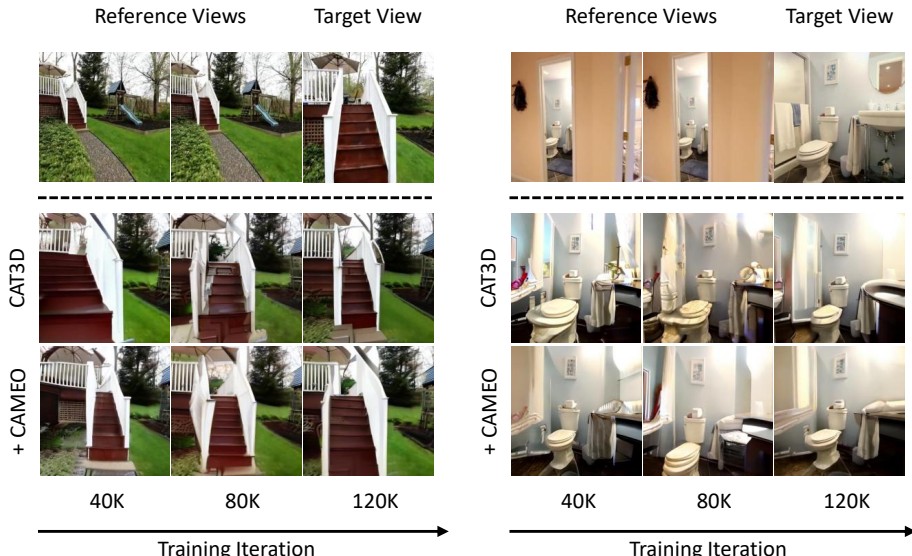

Figure 5: **Qualitative results on RealEstate10K.** CAMEO captures camera poses faster than the baseline, as its explicit correspondence supervision allows the model to learn geometric consistency more efficiently, leading to quicker convergence novel view synthesis.

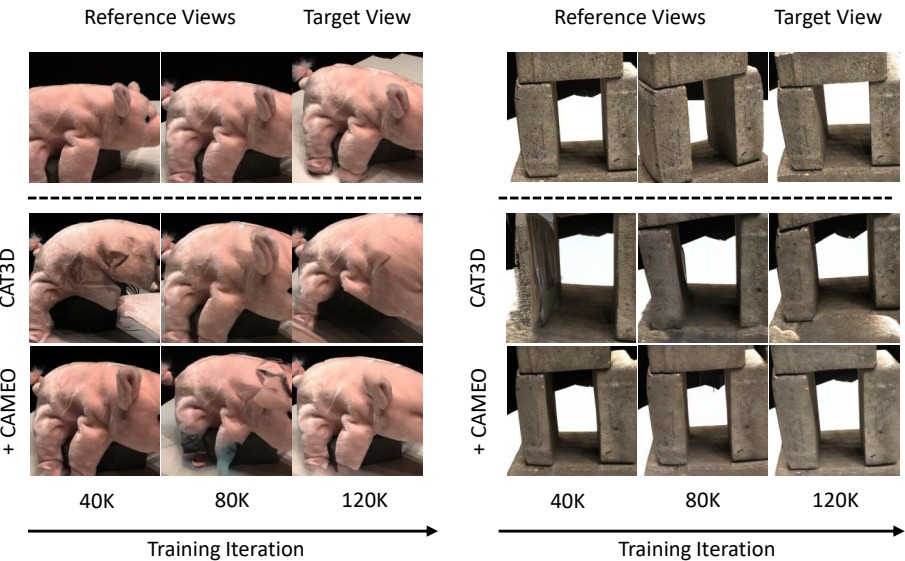

Figure 6: **Qualitative results on DTU.** CAMEO preserves object structure (e.g., pig's legs, bricks structure) better than the baseline, producing more geometrically consistent novel views.

that CAMEO produces novel views that are more aligned with the ground-truth images. Also, as shown in Figure 6, CAMEO better preserves object structure compared to the baseline. These results confirm that explicit correspondence supervision enhances geometric consistency and significantly improves overall NVS quality. Additional qualitative examples are provided in Appendix D.

**Generalization to OOD setting.** The advantages of CAMEO are not limited to in-domain settings. As shown in Table 1, even when evaluated on the object-centric DTU dataset —which differs significantly from the training distribution of RealEstate10K— our method consistently outperforms the baseline. This suggests that CAMEO enables the model to learn a generalizable geometric understanding that extends beyond the training distribution.

Table 1: **Novel View Synthesis Evaluation** on RealEstate10K (Zhou et al., 2018) and DTU (Jensen et al., 2014). Iter. denotes the training iteration.

| Model | Iter. | RealEstate10k | | | DTU (Out-of-distribution) | | |
|---|---|---|---|---|---|---|---|
| | | PSNR ↑ | SSIM ↑ | LPIPS ↓ | PSNR ↑ | SSIM ↑ | LPIPS ↓ |
| CAT3D | | **17.12** | 0.601 | **0.276** | 9.31 | **0.286** | 0.623 |
| CAT3D + REPA (Yu et al., 2024) | 20k | 17.03 | **0.604** | 0.294 | - | - | - |
| **CAT3D + CAMEO** | | 16.89 | **0.604** | 0.290 | **10.01** | 0.275 | **0.592** |
| CAT3D | | 17.32 | 0.611 | 0.274 | 9.90 | 0.294 | 0.613 |
| CAT3D + REPA (Yu et al., 2024) | 40k | 17.55 | 0.618 | 0.262 | - | - | - |
| **CAT3D + CAMEO** | | **17.87** | **0.634** | **0.254** | **10.77** | **0.309** | **0.535** |
| CAT3D | | 18.09 | 0.641 | 0.260 | 10.30 | 0.307 | 0.573 |
| CAT3D + REPA (Yu et al., 2024) | 80k | 17.86 | 0.633 | 0.253 | - | - | - |
| **CAT3D + CAMEO** | | **18.34** | **0.646** | **0.245** | **11.45** | **0.373** | **0.510** |
| CAT3D | | 18.06 | 0.640 | 0.243 | 11.24 | 0.352 | 0.532 |
| CAT3D + REPA (Yu et al., 2024) | 120k | 18.17 | 0.644 | 0.246 | - | - | - |
| **CAT3D + CAMEO** | | **18.65** | **0.659** | **0.233** | **11.54** | **0.366** | **0.523** |
| CAT3D | 160k | 18.30 | 0.644 | 0.246 | - | - | - |

Table 2: **Ablation studies of CAMEO** on the RealEstate10K dataset (Zhou et al., 2018). All experiments use 40k training iterations.

| Layer | Iter. | PSNR ↑ | SSIM ↑ | LPIPS ↓ |
|---|---|---|---|---|
| 4 | 40k | 18.61 | 0.648 | 0.227 |
| 8 | 40k | 18.51 | 0.645 | 0.222 |
| **10** | 40k | **18.77** | **0.658** | **0.216** |

(a) Ablation on alignment layer

| $\lambda$ | Iter. | PSNR ↑ | SSIM ↑ | LPIPS ↓ |
|---|---|---|---|---|
| 0.01 | 40k | 18.44 | 0.643 | 0.228 |
| **0.02** | 40k | **18.56** | **0.655** | **0.223** |
| 0.04 | 40k | 18.46 | 0.648 | 0.232 |

(c) Ablation on loss weight ($\lambda$)

| MLP head | Iter. | PSNR ↑ | SSIM ↑ | LPIPS ↓ |
|---|---|---|---|---|
| × | 40k | **18.77** | 0.651 | 0.226 |
| ✓ | 40k | **18.77** | **0.658** | **0.216** |

(b) Ablation on MLP head

| Loss function | Iter. | PSNR ↑ | SSIM ↑ | LPIPS ↓ |
|---|---|---|---|---|
| L1 | 40k | 18.55 | 0.652 | 0.234 |
| **Cross entropy** | 40k | **18.77** | **0.658** | **0.216** |

(d) Ablation on loss function

## 4.3 ABLATION STUDY

To analyze the contribution of each component in CAMEO, we conduct ablation studies by varying its core modules, including the alignment layer, the presence of an MLP head, the weighting parameter $\lambda$, and the choice of loss function. All experiments are performed on the RealEstate10K test set (Zhou et al., 2018) with 40k training iterations per setting. As shown in Table 2, our ablation study shows that using layer $\ell = 10$ yields the best performance, which agrees with our earlier analysis.

## 5 CONCLUSION

We present CAMEO, a simple framework to improve multi-view diffusion models. Our work is the first to reveal that the model's inflated 3D attention maps inherently capture geometric correspondences across views. Furthermore, we demonstrate that this property is strongly correlated with novel view synthesis quality. Building on these findings, CAMEO introduces geometric supervision into a model's attention layer, guiding the model to learn more precise geometry feature. This leads to faster convergence and improved generation quality. CAMEO is model-agnostic and can be integrated into any multi-view diffusion architecture that employs inflated 3D attention. We hope our findings inspire further research in geometry-aware generative modeling, including applications in attention distillation. Further discussion is provided in Appendix E.

## REPRODUCIBILITY STATEMENT

We detail the training configurations in Section 4.1 and Section 4.2. We will also release our code and model checkpoints to ensure reproducibility.

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

# APPENDIX

In this appendix, Section A reviews the fundamentals of diffusion models. Section B describes the architecture of multi-view diffusion models in detail. Section C presents additional quantitative and qualitative analyses. Section D provides further qualitative results and Section E reports limitations and future work. Lastly, Section F documents the use of LLMs in our work.

## A    DESCRIPTORS FOR DIFFUSION MODELS

Diffusion models(Ho et al., 2020; Song et al., 2020) are a class of generative models that learn data distributions by reversing a gradual noising process. Starting from clean data samples $x_0 \sim p_{\text{data}}(x)$, a forward process incrementally corrupts them with Gaussian noise to produce a sequence of latent variables $\{x_t\}_{t=1}^T$. A neural network is then trained to approximate the reverse process, progressively denoising a sample from pure Gaussian noise back into a realistic data point.

### A.1    DENOISING DIFFUSION PROBABILISTIC MODELS

Denoising Diffusion Probabilistic Models (DDPM) (Ho et al., 2020) define a forward noising process $q(x_t|x_{t-1})$ with a variance schedule $\{\beta_t\}_{t=1}^T$, where $\alpha_t = 1 - \beta_t$ and $\bar{\alpha}_t = \prod_{s=1}^t \alpha_s$. At an arbitrary timestep $t$, the closed form of the noising process is

$$x_t = \sqrt{\bar{\alpha}_t}\, x_0 + \sqrt{1 - \bar{\alpha}_t}\, \epsilon, \quad \epsilon \sim \mathcal{N}(0, I). \tag{7}$$

The generative task is to learn the reverse process $p_\theta(x_{t-1}|x_t)$ such that a sample from $x_T \sim \mathcal{N}(0, I)$ can be gradually denoised to yield $x_0 \sim p_{\text{data}}$. In practice, this reverse transition is parameterized by a neural network $\epsilon_\theta(x_t, t)$ that predicts the noise, leading to

$$p_\theta(x_{t-1}|x_t) := \mathcal{N}\left(x_{t-1}; \frac{1}{\sqrt{\alpha_t}}\left(x_t - \frac{\beta_t}{\sqrt{1 - \bar{\alpha}_t}}\, \epsilon_\theta(x_t, t)\right), \sigma_t^2 I\right), \tag{8}$$

where $\sigma_t^2$ can be fixed or learned. Training is performed with the denoising objective

$$\mathcal{L}_{\text{simple}}(\theta) = \mathbb{E}_{x_0, \epsilon, t}\left[\|\epsilon - \epsilon_\theta(x_t, t)\|_2^2\right], \tag{9}$$

which corresponds to score matching (Hyvärinen & Dayan, 2005), since $\epsilon_\theta(x_t, t)$ approximates the score function $-\sigma_t \nabla_{x_t} \log p(x_t)$. Moreover, by reparameterization one can directly obtain an estimate of the clean sample $x_0$ at timestep $t$ as

$$\hat{x}_0(x_t) = \frac{1}{\sqrt{\bar{\alpha}_t}}\left(x_t - \sqrt{1 - \bar{\alpha}_t}\, \epsilon_\theta(x_t, t)\right), \tag{10}$$

which provides an explicit reconstruction of the data from noisy inputs and plays a key role in both DDPM sampling and extensions such as DDIM.

### A.2    DENOISING DIFFUSION IMPLICIT MODELS

Denoising Diffusion Implicit Models (DDIM) (Song et al., 2020) build upon DDPM but modify the formulation to allow for a deterministic, non-Markovian sampling procedure that substantially accelerates generation. Instead of requiring $T$ iterative reverse steps, DDIM introduces a reparameterized reverse process where the current latent $x_t$ can be deterministically mapped to $x_{t-1}$ using both the predicted clean image $\hat{x}_0(x_t)$ and the predicted noise $\epsilon_\theta(x_t, t)$. Specifically, the reverse update is

$$x_{t-1} = \sqrt{\bar{\alpha}_{t-1}}\, \hat{x}_0(x_t) + \sqrt{1 - \bar{\alpha}_{t-1}}\, \epsilon_\theta(x_t, t). \tag{11}$$

This deterministic formulation allows one to skip intermediate steps in the reverse trajectory without retraining the model, leading to fast sampling while preserving high generative quality. DDIM thus serves as a practical alternative to DDPM and is widely adopted in applications such as Stable Diffusion, where efficient and scalable generation is crucial.

# B  DETAILS OF MULTI-VIEW DIFFUSION MODEL

Figure 7: **Model architecture of CAT3D (Gao et al., 2024)**

**Architecture Overview.**  Our baseline model is CAT3D (Gao et al., 2024), a multi-view extension of Stable Diffusion 2.1 (Rombach et al., 2022). CAT3D adapts the latent text-to-image diffusion framework by inflating the 2D self-attention layers into 3D self-attention, enabling interactions across different views. Although the official implementation and model weights of CAT3D are not publicly available, we adopt the reproduction provided by MVGenMaster (Cao et al., 2025), which faithfully replicates CAT3D's training and evaluation pipeline.

**Network Structure.**  The underlying architecture consists of three downsampling blocks, one mid-block, and three upsampling blocks. Each downsampling block contains two layers, the mid-block contains one layer, and each upsampling block contains three layers. Each layer comprises a spatial convolution followed by a self-attention module.

In CAT3D, standard self-attention layers are replaced with inflated 3D self-attention layers to capture inter-view dependencies. This 3D attention is applied in all blocks except the first and last (i.e., it is implemented in downsampling blocks 2 & 3, the mid-block, and upsampling blocks 1 & 2). In total, there are 11 inflated 3D self-attention layers used in our analysis.

The input images of resolution $512 \times 512$ are encoded by the VAE encoder into latent features of size $64 \times 64$. Gaussian noise is added to the target latents for generation, while the reference latents remain unchanged. To form the conditioning latent, we first compute the Plücker ray embedding (Xu et al., 2023), which encodes per-pixel camera rays, and concatenate it with a binary visibility mask indicating the reference images. This conditioning signal is then passed through a shallow convolutional network to match the dimensionality of the image latents. Finally, the conditioning latents are added to the image latents, producing the multi-view input representation for the diffusion U-Net.

Each downsampling block reduces the spatial resolution by a factor of 2, producing feature maps of size $32 \times 32$, $16 \times 16$, and $8 \times 8$, respectively. The mid-block operates at the lowest resolution of $8 \times 8$. The upsampling blocks then progressively restore the spatial resolution back to $16 \times 16$, $32 \times 32$, and $64 \times 64$. Finally, the latent is passed through the VAE decoder to reconstruct the full-resolution image of size $512 \times 512$.

Table 3: **Layer-wise Pearson correlation coefficient ($r$) and p-values ($p$) between alignment error and perceptual metrics.** Signs for PSNR and SSIM are inverted so that a higher $r$ value consistently means better. Statistical significance is reported with p-values: **bold** indicates $p < 0.01$ and underline indicates $0.01 \leq p < 0.05$.

| | LPIPS | | PSNR | | SSIM | |
|---|---|---|---|---|---|---|
| Layer | $r$ | $p$ | $r$ | $p$ | $r$ | $p$ |
| $l = 2$ | **0.60** | **1.60e-18** | **0.29** | **1.16e-3** | **0.27** | **1.83e-3** |
| $l = 3$ | **0.60** | **7.25e-20** | **0.29** | **6.61e-4** | **0.25** | **1.53e-3** |
| $\ell = 4$ | **0.53** | **1.07e-14** | 0.23 | 1.72e-2 | 0.02 | 6.09e-1 |
| $\ell = 5$ | **0.53** | **2.46e-11** | 0.26 | 3.23e-2 | 0.06 | 4.42e-1 |
| $\ell = 6$ | -0.13 | 1.07e-1 | -0.11 | 1.20e-1 | 0.11 | 1.95e-1 |
| $\ell = 7$ | **0.67** | **4.27e-20** | **0.42** | **2.93e-5** | 0.15 | 1.78e-1 |
| $\ell = 8$ | **0.60** | **6.01e-16** | **0.32** | **1.02e-3** | 0.05 | 3.98e-1 |
| $\ell = 9$ | **0.65** | **7.46e-19** | **0.40** | **6.33e-4** | 0.12 | 2.72e-1 |
| $\ell = 10$ | **0.73** | **2.92e-29** | **0.50** | **8.99e-9** | **0.32** | **8.17e-4** |
| $\ell = 11$ | **0.73** | **4.24e-29** | **0.50** | **2.58e-8** | **0.27** | **4.27e-3** |
| $\ell = 12$ | **0.72** | **2.53e-28** | **0.50** | **1.39e-7** | **0.28** | **7.68e-3** |

## C  FURTHER ANALYSIS

### C.1  DETAILED ANALYSIS SETUP

For our analysis, we trained and validated our model on the RealEstate10K (Zhou et al., 2018) dataset using the equivalent settings as described in Section 4. We used checkpoints saved at 2k, 4k, 8k, 16k, 32k, and 64k training iterations. Specifically, the results presented in Figure 3(a) of the main paper are based on the 64k checkpoint. The correlation analysis and alignment error measurements were conducted on the first 200 scenes of the validation set, using two reference views and one target view for each scene. The following sections present supplementary results and deeper quantitative and qualitative analysis of our findings.

### C.2  DETAILED QUANTITATIVE ANALYSIS

**Statistical Significance of Pearson Correlation Coefficient.**  Table 3 presents the statistical significance of the correlations shown in Figure 3. We calculated the p-value for each correlation to assess its statistical validity. The results indicate that at a significance level of $p < 0.01$, all layers in the Up 2 block ($\ell = 10, 11, 12$) exhibit a statistically significant correlation. Among these, layer $\ell = 10$ demonstrated the highest effect size ($r$), confirming it as the most critical layer for geometric correspondence.

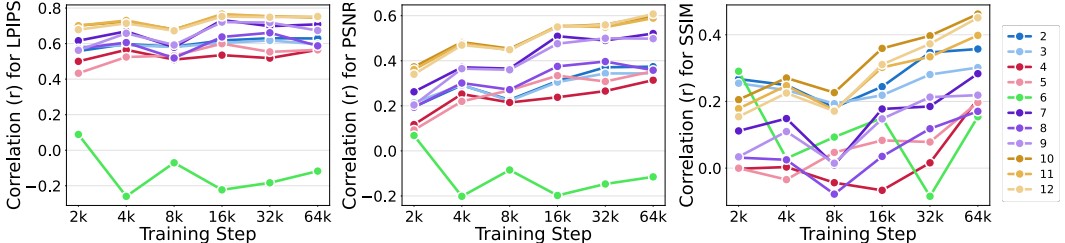

Figure 8: **Pearson correlation coefficient between $\mathcal{E}_{\text{align}}^{l}$ and perceptual metrics across training iterations.**

**Correlation across training iterations.**  Figure 8 illustrates how the Pearson correlation coefficient between the alignment error ($\mathcal{E}_{\text{align}}^{\ell}$) and perceptual metrics evolves throughout the training process. The effect size $r$ for almost every layer and metric strengthened as training progressed.

The mid-layer ($\ell = 6$) was the only one that did not follow this pattern. The final 64k iteration checkpoint, which exhibits the strongest correlations, corresponds to the results presented in Figure 3(a).

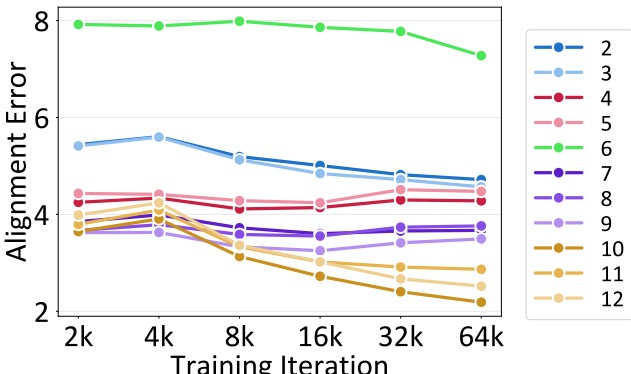

Figure 9: $\mathcal{E}_{\text{align}}^{\ell}$ **across training iterations at different layers.**

**Alignment Error Across All Layers.** We report the alignment error $\mathcal{E}_{\text{align}}^{\ell}$ between the attention maps and ground-truth correspondences across training iterations for all layers (Figure 9). Layers within the same block exhibit similar trends: Down 2 ($\ell = 2, 3$), Down 3 ($\ell = 4, 5$), Mid ($\ell = 6$), Up 1 ($\ell = 7, 8, 9$), and Up 2 ($\ell = 10, 11, 12$).

Layers in Up 2 ($\ell = 10, 11, 12$) achieve the best alignment, starting with the smallest error and further reducing it during training. Down 3 and Up 1 ($\ell = 4, 5, 7, 8, 9$) begin with relatively low error but maintain their initial values. Down 2 ($\ell = 2, 3$) shows some reduction, but the error remains high in absolute terms. The mid-layer ($\ell = 6$) consistently exhibits the largest error.

Overall, Up 2 layers align most strongly with the correspondence maps, with layer $\ell = 10$ showing the lowest final error. This suggests an emergent property: certain layers adapt their attention maps to encode geometric correspondence information.

**Alignment Error per Attention Head.** Attention layers are composed of multiple heads, where each head operates on a different subspace of the feature representation (Vaswani et al., 2017). This design allows different heads to capture diverse types of relationships, such as local structure or long-range dependencies. To analyze this behavior of multi-view diffusion models, we examine the attention alignment of individual heads. As shown in Figure 10, within the same layer, the absolute values vary across heads, but the overall trends remain consistent within each block. Therefore, for both analysis and experiments, we report the results averaged over all heads.

### C.3 DETAILED QUALITATIVE ANALYSIS

We visualize attention maps across different layers of the model. Since the spatial dimensions of the attention maps vary by layer, the resolution of the visualizations also differs. Notably, attention layers from Up 2 Block produce patterns that closely resemble the geometric correspondence maps. In particular, the attention map of $\ell = 10$ captures accurate geometric correspondences, successfully identifying the counterpart of a query point in another view.

Another key observation is that some layers encode semantic correspondences rather than purely geometric ones. For instance, in Figure 12b, the attention map at layer $\ell = 4$ shows that when the query is placed on the right side of a chair, the attention also highlights the chair's left side. This indicates that the layer is responsible for capturing semantic counterparts. Overall, these findings suggest that different layers specialize in distinct types of information, with some focusing on geometry and others on semantics.

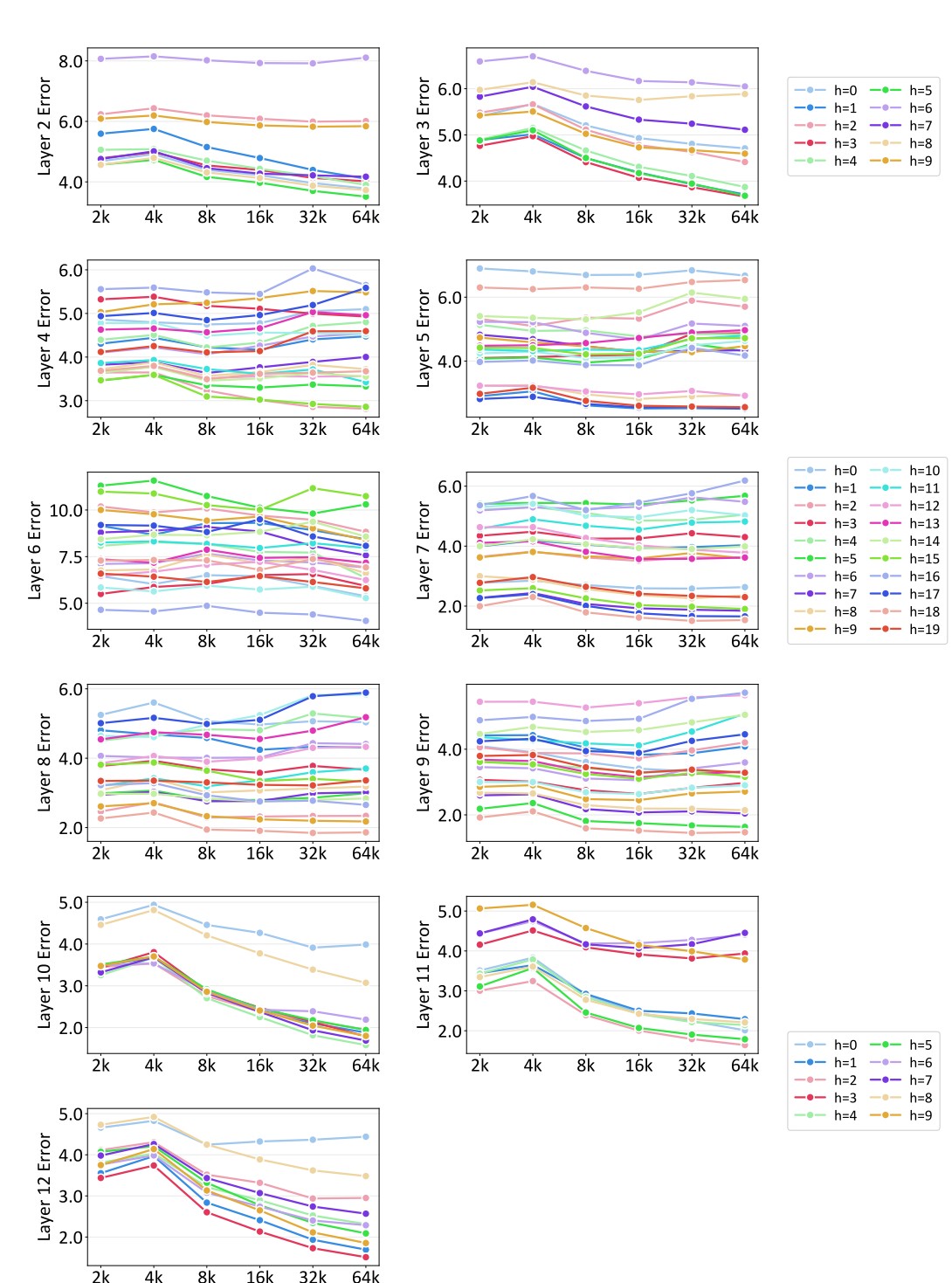

Figure 10: $\mathcal{E}_{\text{align}}^{\ell}$ **across training iterations for different heads in layer** $\ell$.

## D  MORE QUALITATIVE RESULTS

This section provides additional qualitative comparisons of CAMEO on both scene-level (Zhou et al., 2018) and object-centric (Jensen et al., 2014) settings. We present visual comparisons against the baseline, highlighting improved geometric consistency. Figure 11 shows qualitative examples organized by training iteration.

## E  LIMITATIONS AND FUTURE WORK

- **Hyperparameter sensitivity.** Our method is sensitive to hyperparameters, such as the learning rate and the distillation weight. Careful tuning is required for stable training and optimal performance. Future work could explore adaptive or automated strategies to reduce this sensitivity.

- **Dependence on external geometry.** CAMEO relies on point clouds obtained from an external geometry estimation model (Wang et al., 2025) to construct geometric correspondence maps. Reducing this dependence or jointly learning geometry and diffusion would make the framework more broadly applicable.

- **Beyond novel view synthesis.** Our method targets multi-view diffusion for novel view synthesis. Extending correspondence-aware supervision to video diffusion, 4D reconstruction, or other multi-modal tasks remains an open direction.

- **Beyond U-Net architectures.** Our experiments are conducted on U-Net based multi-view diffusion models (Gao et al., 2024). Extending CAMEO to DiT-based architectures (Peebles & Xie, 2023) would be an important step toward demonstrating that correspondence-aware supervision is a universal framework for multi-view diffusion.

## F  THE USE OF LARGE LANGUAGE MODELS(LLMS)

We employed large language models (LLMs) in two parts of our workflow:

- **Literature exploration.** We used LLMs to suggest potentially relevant works during the initial research phase. However, all cited papers in our related work section were manually reviewed to ensure their relevance and accuracy before inclusion.

- **Writing assistance.** We used LLMs to help refine our writing, including correcting grammar, improving clarity, and formatting mathematical expressions. All technical content and claims were authored and verified by the authors.

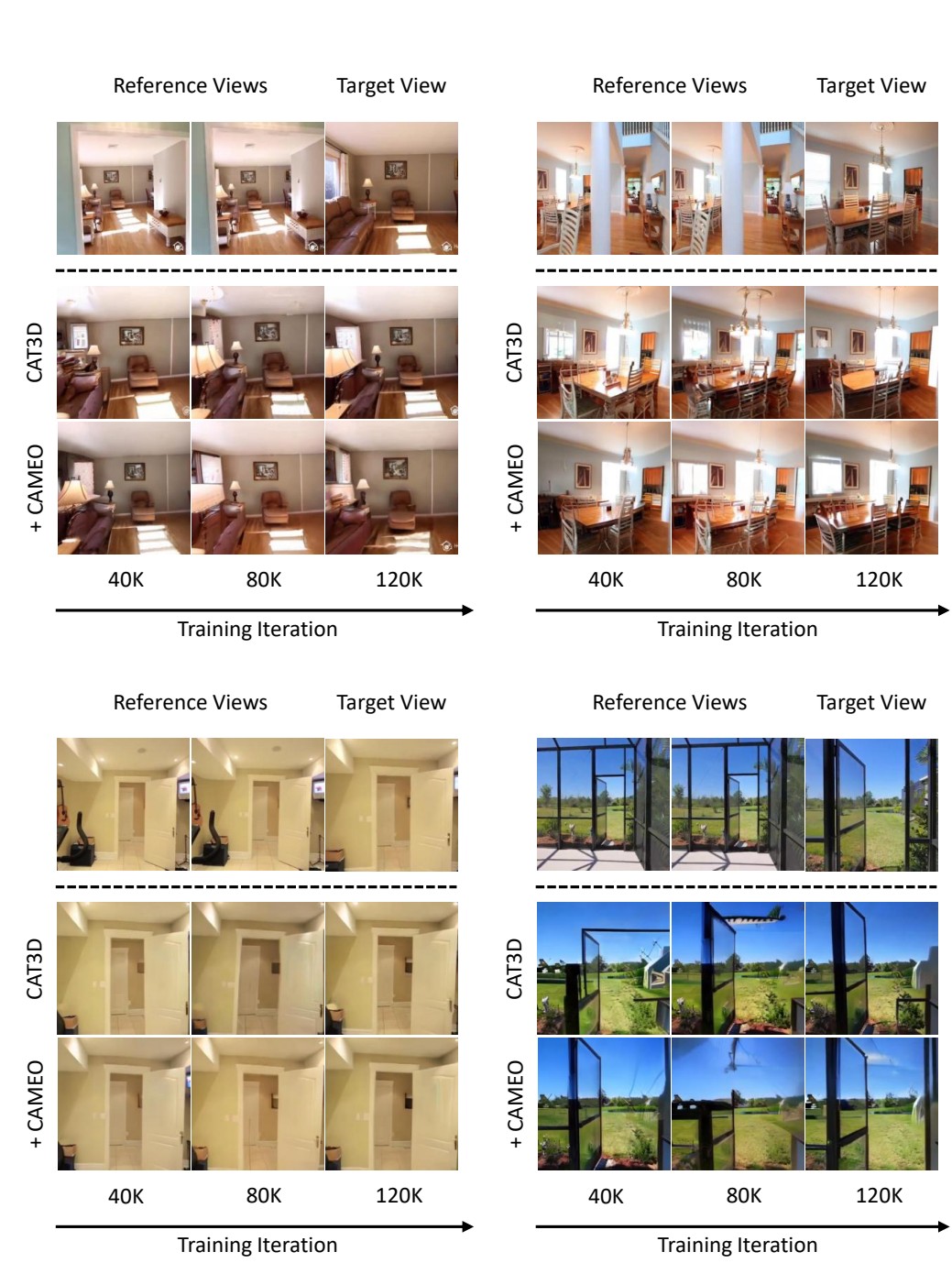

Figure 11: **Additional qualitative results.** Each panel corresponds to a different sample; within each panel, results are organized by training iteration.

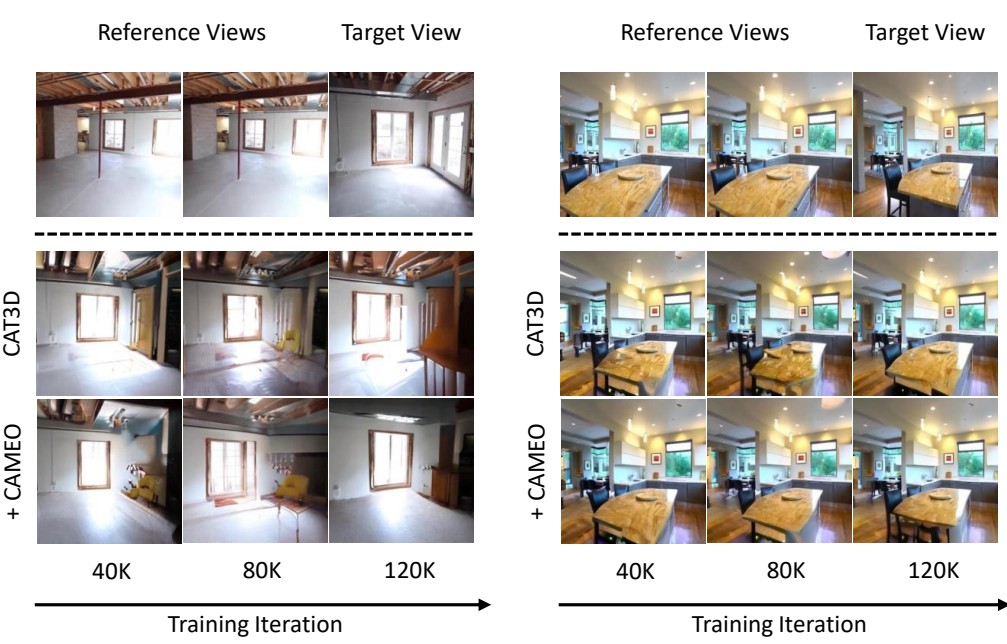

Figure 11: **Additional qualitative results on benchmark datasets (continued).**

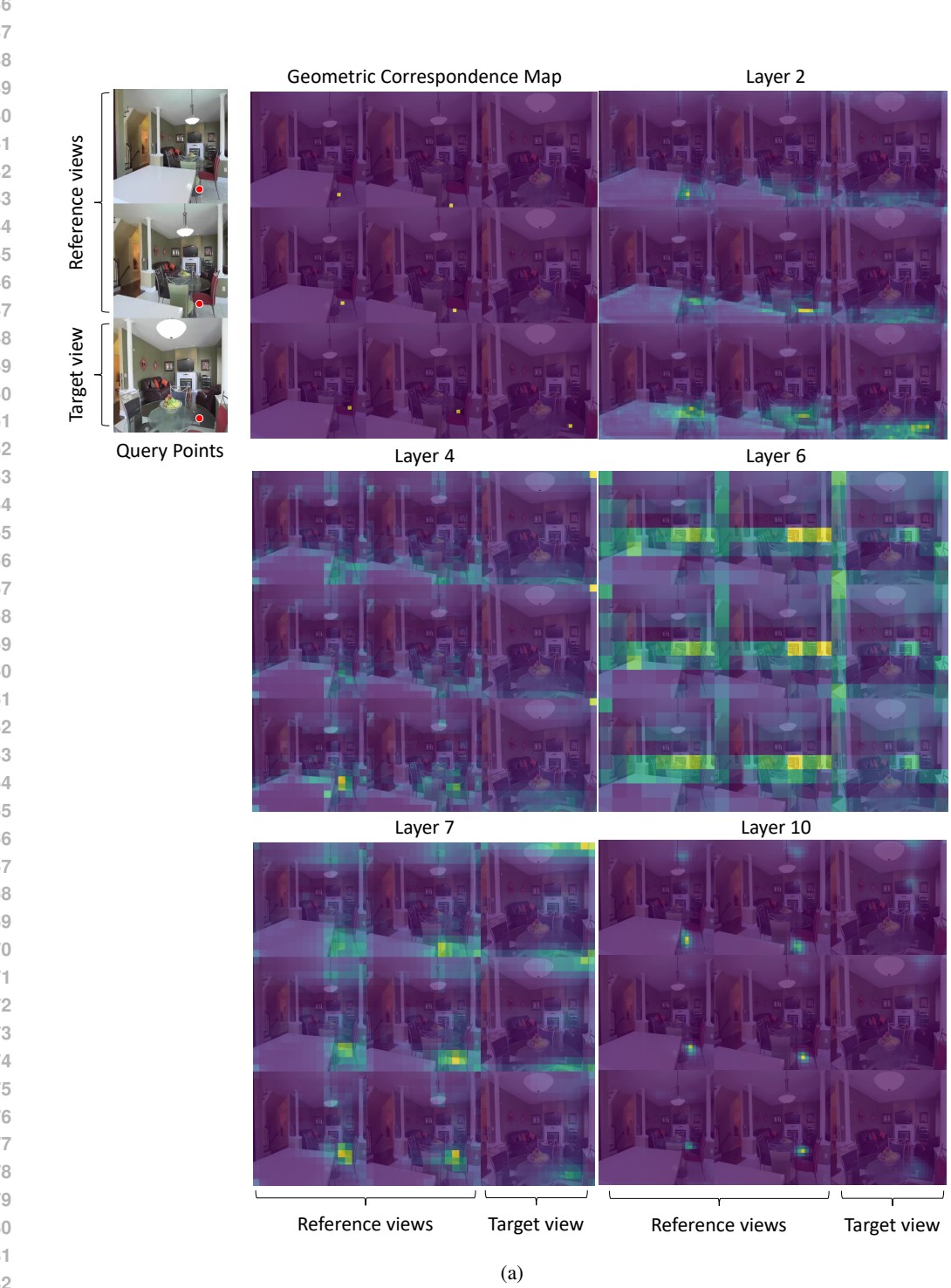

(a)

Figure 12: **Visualization of inflated 3D attention maps across different layers of the model.**

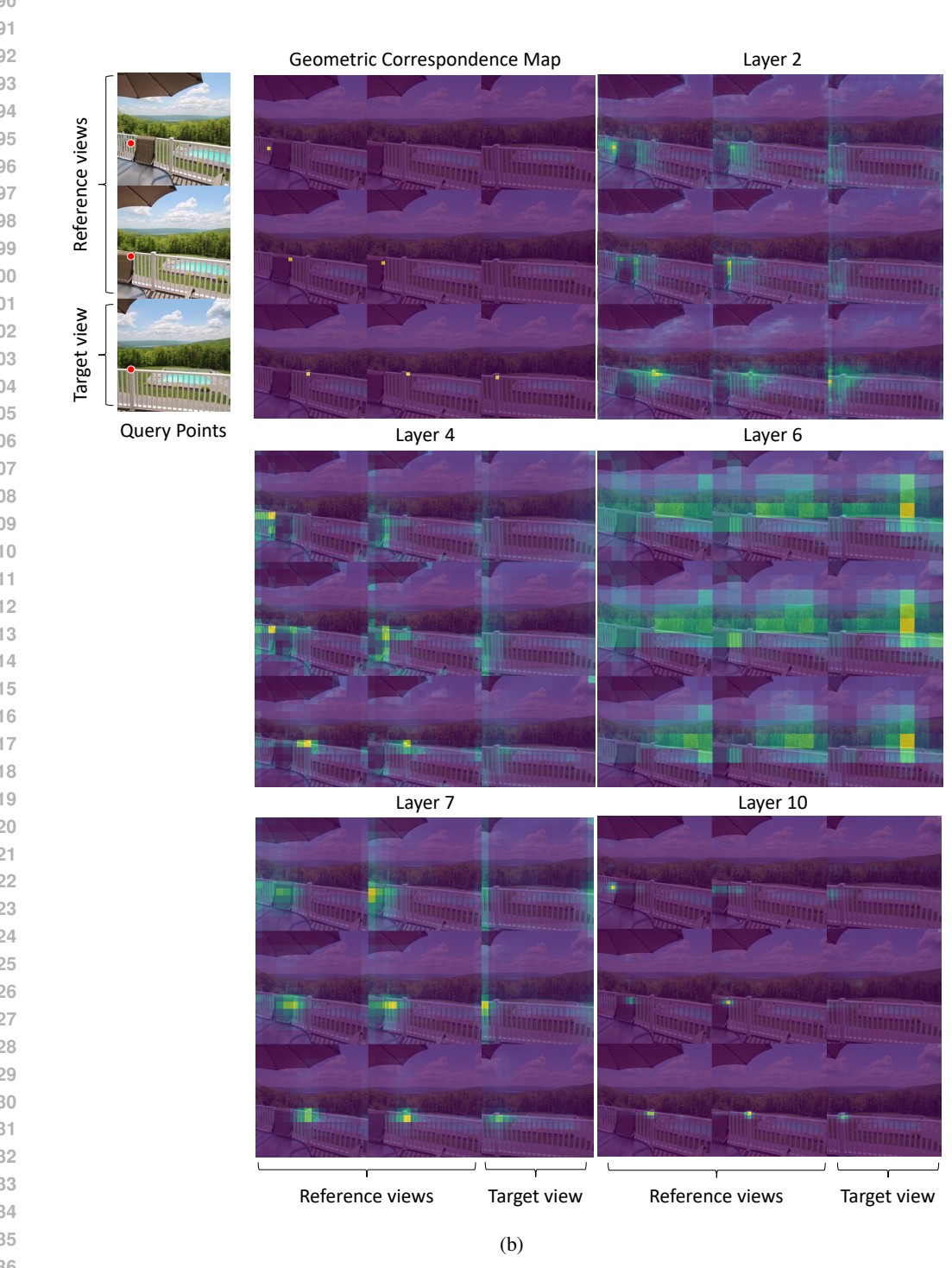

Figure 12: **Visualization of inflated 3D attention maps across different layers of the model (continued).**

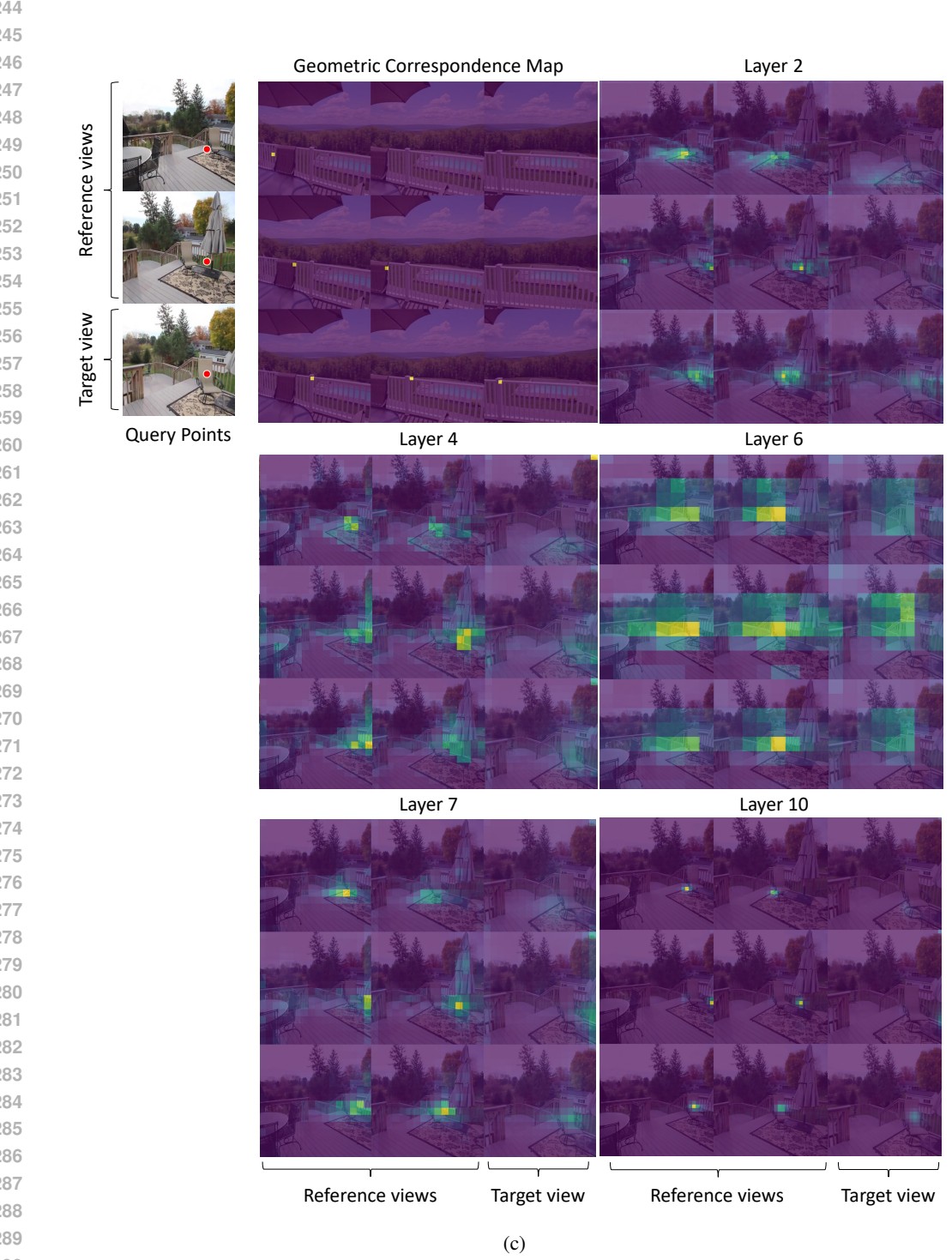

Figure 12: **Visualization of inflated 3D attention maps across different layers of the model (continued).**

