# OpenReview forum: "CAMEO: Correspondence-Attention Alignment for Multi-View Diffusion Models"
_ICLR.cc/2026/Conference — ICLR 2026 Conference Withdrawn Submission_

### Official Review · Reviewer_N1AH · 2025-10-27

**Soundness:** 2
**Presentation:** 4
**Contribution:** 3
**Rating:** 4
**Confidence:** 4

**Summary:**

The paper investigates the geometric correspondence problem of multi-view diffusion models in novel view synthesis (NVS) task.
First, it introduces alignment error, a metric quantifying how well attention maps capture inter-view correspondences between target and reference images, compared to ground-truth correspondence maps derived from 3D point maps.

Second, the authors find that final attention layers in the U-Net architecture exhibit lower alignment error. Furthermore, this metric correlates
strongly with standard image quality scores (PSNR, SSIM, LPIPS), validating its effectiveness.

Finally, the paper proposes to explicitly supervise geometric correspondence by incorporating alignment error as a regularization term during training. This simple yet effective strategy leads to faster convergence and higher-quality novel views, as demonstrated in experiments.

**Strengths:**

* The paper is well-written and well-structured, with clear language and well-designed figures that make the methodology and results easy to follow and understand. Overall, the editorial quality of this work is very good, and helpful.

* The authors investigate an important problem: the geometric consistency in novel view synthesis (NVS) task. The proposed idea (supervising attention maps using geometric correspondence signals) is simple yet effective.

* The analysis of the validity of the proposed alignment metric is thorough and clearly presented. I value the correlation analysis between the
proposed metric at the final layers and standard image quality metrics, as well as the investigation of its evolution during training across different numbers of iterations.

**Weaknesses:**

* The novelty of the paper is limited, as the core method is primarily based on a combination and refinement of existing techniques, such as selecting cross-attention layers that best capture the target semantics, the use of an off-the-shelf pretrained model to obtain ground-truth semantic maps, and taking the discrepancy with the ground truth as a regularization loss to refine attention units and improve semantic consistency. The paper uses and adapts these existing ideas to the novel view synthesis (NVS) task.

* The analysis about generality and scalability of the proposed approach across different model architectures and datasets require further investigation. The paper experimented only on one U-Net-based baseline model (CAT3D) and primarily on one dataset (RealEstate10K) for both metric analysis and main evaluation, with only one additional dataset used for the OOD task. The key findings that final attention layers better capture geometric correspondences and that the alignment loss should be applied at layer 10 may be specific to this particular architecture and dataset. It remains unclear whether these findings and design choices would remain robust and consistent on other architectures and datasets. Evaluating the method on DiT-based diffusion models and on multiple datasets would better demonstrate robustness and scalability. For reference, the CAT3D paper experimented on four datasets, and the REPA paper tested multiple DiT-based models at various sizes.

* The paper considers only one competitor (REPA) in addition to the baseline model, which is insufficient for a comprehensive evaluation, especially given that REPA is not specifically designed to address geometric consistency in the novel view synthesis (NVS) task.
In addition, to ensure a fair comparison, the paper should explicitly describe necessary implementation details of REPA, including the weight of the representation alignment loss and the layers to which this loss was applied during training.

* In some of the qualitative results, the proposed method appears to degrade image quality. For example, in Figure 5 at Training Iteration
= 120K, using CAMEO makes the staircase more blurred in the left example and causes the mirror to disappear in the right
example.

**Questions:**

* Generalizability across architectures: Would the authors consider testing CAMEO on DiT models at different sizes (e.g. the B/2 and L/2
architectures introduced in the DiT paper)? It would be very helpful to understand the design robustness under different modeling choices.

* Configuration of REPA: What are the implementation details of REPA? For example, what is the weight of the representation alignment loss? To which layers was this loss applied during training? What external pretrained visual encoder was used to compute the REPA loss?

* Training iterations: In Figure 4(a), although CAMEO shows faster convergence, the training curves for all three methods have not yet fully
converged. If training were continued until all methods reach convergence, how would their final performances compare? Additionally, in
Table 1, why was only CAT3D trained for 160k iterations?

---

### Official Review · Reviewer_XWQm · 2025-10-30

**Soundness:** 2
**Presentation:** 2
**Contribution:** 2
**Rating:** 2
**Confidence:** 5

**Summary:**

This paper proposes CAMEO, a simple attention–correspondence alignment method for multi-view diffusion models. It explicitly supervises attention maps with geometric correspondence signals derived from 3D point maps, improving training efficiency and novel view synthesis quality. Applied to CAT3D, CAMEO achieves ~2× faster convergence and slightly better PSNR/SSIM results on RealEstate10K and DTU.

**Strengths:**

Clear observation that attention maps naturally encode geometric correspondences.

Demonstrates improvement and faster convergence over CAT3D.

**Weaknesses:**

The correspondence-alignment idea is simple and not very novel.

Depends on external geometry estimation, need to preprocess the training data which is not scalable. Its unclear how much the computational cost to get the correspondence.

Only marginal final quality improvement over CAT3D, and the claimed 2x training speedup is measured by PSNR, which is not really reliable when its below 20.

Limited evaluation on one backbone; lacks broader comparison with SoTA.

Reliability and scalability of correspondence maps not fully analyzed.

**Questions:**

Please see the weakness section.

---

### Official Review · Reviewer_mYz4 · 2025-10-30

**Soundness:** 2
**Presentation:** 2
**Contribution:** 2
**Rating:** 2
**Confidence:** 5

**Summary:**

This paper analyzes the attention maps of multi-view diffusion models and shows they correlated with geometric correspondence, even though these models are only trained with a 2D denoising loss and have no explicit 3D inductive bias. Based on this observation, the authors propose CAMEO, an attention-alignment method that explicitly aligns multi-view attention with geometric correspondenses derived from 3D point maps. Built on the CAT3D framework, CAMEO aims to accelerate converagence and improve quality.

**Strengths:**

1. The paper is cearly written.
2. The architecture of CAMEO is simple that it just adds an auxiliary cross-entropy loss.

**Weaknesses:**

1. The claim that inflated 3D attention "emerges" as geometric correspondence feels incremental. Prior work has already observed that the attention map encodes point-to-point or identity-consistent correspondences in diffusion models [1,2]. Extending this observation to multi-view NVS is interesting but not obviously surprising.
2. The paper claims that CAMEO achieves 2x faster convergence. However, this is not clearly supported by the results in Figure 1: the LPIPS curves do not exhibit such a speedup as seen in PSNR, and around 60k iterations, the PSNR difference between CAMEO and the baseline is marginal. Beyond that point, the reported convergence speed improvements seem to reflect that the baseline has already converged, rather than indicating that CAMEO is fundamentally faster.
3. Although the method itself is simple, the supervision pipeline is not. Generating the correspondence maps requires estimating per-pixel 3D point maps for each view pair using an external geometry estimator, then performing point matching in 3D and filtering with cycle-consistency. This adds cost and relies on the accuracy of the estimator. The paper does not quantify that overhead.
4. The choice of which attention layer to supervise is manually selected, based on empirical correlation with geometric alignment and PSNR. However, the robustness of this choice across architectures is unclear. In text-to-image tasks, many prior works [3,4] have shown that attention behavior varies with model architecture and layer depth, and can be unstable when transferred across models. This suggests that CAMEO’s reliance on a specific layer may not generalize well and could require per-architecture tuning.  This raises questions about how "model-agnostic" the method really is.
5. The cycle-consistency mask uses a threshold $\tau$ to drop unreliable matches, and this mask gates which correspondences supervise the attention. The paper does not present a sensitivity analysis on $\tau$.

[1] Tang, Luming, et al. "Emergent correspondence from image diffusion." Advances in Neural Information Processing Systems 36 (2023): 1363-1389.
[2] Wang, Mengyu, et al. "Characonsist: Fine-grained consistent character generation." Proceedings of the IEEE/CVF International Conference on Computer Vision. 2025.
[3] Cao, Mingdeng, et al. "Masactrl: Tuning-free mutual self-attention control for consistent image synthesis and editing." Proceedings of the IEEE/CVF international conference on computer vision. 2023.
[4] Tumanyan, Narek, et al. "Plug-and-play diffusion features for text-driven image-to-image translation." Proceedings of the IEEE/CVF conference on computer vision and pattern recognition. 2023.

**Questions:**

How robust is the method to the choices of $\tau$?

---

### Official Review · Reviewer_sUqX · 2025-10-31

**Soundness:** 3
**Presentation:** 3
**Contribution:** 3
**Rating:** 4
**Confidence:** 4

**Summary:**

This paper presents CAMEO, a method to improve training efficiency and quality in multi-view diffusion models for novel view synthesis. The authors show that these models implicitly learn geometric correspondences in their inflated 3D attention maps, with alignment in deeper layers correlating to better output metrics like PSNR.

The proposed CAMEO adds a cross-entropy loss to align one attention layer with ground-truth correspondence maps, yielding faster convergence and better NVS results on CAT3D.

**Strengths:**

The work presents how multi-view diffusion models handle geometry without built-in 3D priors, by analyzing its attention behaviors and their link to NVS performance, which is insightful. The method is straightforward, and it addresses a real issue in training efficiency according to the author's claim. Overall, the clarity of paper presentation is good.

**Weaknesses:**

The major weakness of this paper is the insufficient experimental validation.
1. All experiments relies heavily on the model CAT3D, which may limit the evidence of broader applicability. To improve, evaluating on other models (especially another architectures like DiT) would help.
2. While correlations are shown well, establishing causality beyond the proposed fix could use more ablation, such as perturbing attention without alignment.
3. Hyperparameter (e.g., λ) is mentioned but seems to be chosen empirically and the parameter value of the threshold τ in experiments is missing. How did the value selected (λ= 0.02) in the experiments? What is the value of τ in reported experiments?
4. Current related work in this paper covers key areas but overlooks some recent efforts in cross-view alignment for generative NVS, which might provide useful context (refer to *Questions* part).

**Questions:**

I would like to raise several questions about this paper:

1. Robustness:

How does CAMEO perform in challenging cases with sparse views? In particular, what is its robustness to noisier point maps derived from alternative geometry estimation model?

2. Alignment on multiple layers:

Have the authors explored aligning multiple layers, rather than just a single one, in their ablation studies?

3. Generalizability to other architectures:

Has the proposed method been tested on non-U-Net architectures, such as recent DiT-based models?

4. Missing related work:

Several works share a similar philosophy to this one but are absent from the related work section.

[1] Lu et al, 2025. Movis: Enhancing multi-object novel view synthesis for indoor scenes. In Proceedings of the Computer Vision and Pattern Recognition Conference

[2] Li et al, 2025. Nvcomposer: Boosting generative novel view synthesis with multiple sparse and unposed images. In Proceedings of the Computer Vision and Pattern Recognition Conference

In particular, [2] employs an implicit alignment mechanism to distill features from an external model DUSt3R with geometric prior into an generative NVS U-Net, which closely mirrors the approach in this paper.

I would be happy to raise my score if the author rebuttal addresses my concerns.

---

### Note · Authors · 2025-11-12

**Comment:**

We have decided to submit an extended version to another venue.

**Withdrawal Confirmation:**

I have read and agree with the venue's withdrawal policy on behalf of myself and my co-authors.